# In situ observation of thermal-driven degradation and safety concerns of lithiated graphite anode

Xiang Liu [1], Liang Yin[2], Dongsheng Ren[3,4], Li Wang [3,4], Yang Ren [2], Wenqian Xu[2], Saul Lapidus[2], Hewu Wang[4], Xiangming He [3,4], Zonghai Chen [1], Gui-Liang Xu [1✉], Minggao Ouyang [4✉] & Khalil Amine [1,5,6✉]

Graphite, a robust host for reversible lithium storage, enabled the first commercially viable lithium-ion batteries. However, the thermal degradation pathway and the safety hazards of lithiated graphite remain elusive. Here, solid-electrolyte interphase (SEI) decomposition, lithium leaching, and gas release of the lithiated graphite anode during heating were examined by in situ synchrotron X-ray techniques and in situ mass spectroscopy. The source of flammable gas such as $H_2$ was identified and quantitively analyzed. Also, the existence of highly reactive residual lithium on the graphite surface was identified at high temperatures. Our results emphasized the critical role of the SEI in anode thermal stability and uncovered the potential safety hazards of the flammable gases and leached lithium. The anode thermal degradation mechanism revealed in the present work will stimulate more efforts in the rational design of anodes to enable safe energy storage.

[1] Chemical Sciences and Engineering Division, Argonne National Laboratory, Lemont, IL, USA. [2] X-ray Science Division, Advanced Photon Source, Argonne National Laboratory, Lemont, IL, USA. [3] Institute of Nuclear and New Energy Technology, Tsinghua University, Beijing, China. [4] State Key Laboratory of Automotive Safety and Energy, Tsinghua University, Beijing, China. [5] Materials Science and Engineering, Stanford University, Stanford, CA, USA. [6] Institute for Research& Medical Consultations (IRMC), Imam Abdulrahman Bin Faisal University (IAU), Dammam, Saudi Arabia. ✉email: xug@anl.gov; ouymg@tsinghua. edu.cn; amine@anl.gov

With increasing deployment of higher energy density lithium-ion batteries (LIBs) for portable electronic devices and electric vehicles, critical questions surrounding their battery cycle life and safety hinder their large-scale implementation[1,2]. One of the major trade-offs coexisting with increasing battery energy density are concerns about safety[3,4]. To tackle this issue, intensive efforts have focused on the cathode, such as understanding thermal degradation[4–6] and developing countermeasures[7–9], whereas, the thermal degradation pathway and mitigation methods regarding the highly reactive lithiated anode remain elusive. As a result, an adequate practice that can inherently ease the battery safety concern is still missing.

Graphite remains the major anode choice for LIBs because of its overall superior performance, such as high capacity (372 mAh/g), low working potential (~0.1 V vs. Li/Li$^+$), cost-effectiveness, and long cycle life[10,11]. During the charge/discharge process, Li$^+$ ions are inserted/extracted between the graphene planes without significantly disturbing the graphite host structure, thereby achieving reversible (de)intercalations. The success of graphite anodes enabled the first commercially viable LIBs in the early 1990s by outperforming lithium metal anodes in regard to safety and cycling performance, resulting in the nomination for a 2019 Nobel Prize[12,13]. However, many of the phenomena and mechanisms regarding the thermal stability of lithiated graphite remain unclearly understood, such as the stability of lithiated graphite and the solid-electrolyte interphase (SEI), as well as gas release by the anode under thermal-driven forces, etc[14–16].

The thermal stability of lithiated graphite, such as $LiC_{18}$, $LiC_{12}$, and $LiC_6$, are barely understood and even remain controversial[17–19]. For example, lithiated graphite was blamed for reacting with oxygen released by the cathode[20] and even the nonflammable electrolyte[21], thus triggering catastrophic battery runaway failures. However, direct evidence from in situ observations of those reactions and the underlying mechanisms remains lacking. In addition, by means of the density functional theory (DFT) approach Pande et al. calculated that formation of $LiC_6$ is thermodynamically stable and not affect much with 76 K temperature increasing[22]. Moreover, experimental characterizations by means of differential scanning calorimetry (DSC)[23] and X-ray diffraction (XRD)[17], earlier work of Drue[17] and Avdeev et al.[23] also suggested that the $LiC_6$ is thermodynamically stable up to 250–330 °C. Recent work by Andersson[24], Dahn[25], and Aurbach et al.[19] showed, however, that $LiC_6$ starts to decompose at ~80 °C. Such a striking disparity between those results calls for an urgent revisit, especially in an in situ manner to catch any metastable states that would adequately explain the failure modes[26].

On the SEI side, Winter described the SEI as "the most important and the least understood solid electrolyte in rechargeable Li batteries"[27], the SEI thermal evolution is also barely known[15,28]. However, the idea that SEI decomposition accounts for the onset of battery self-heating is well accepted[3]. Due to the lack of in situ characterization tools, its evolution under thermal degradation has never been directly monitored.

Last but not the least, swelling and gas species are prevalent in LIBs[29,30]. Among them, flammable gas, such as hydrogen ($H_2$), has been extensively observed[30–32]. Nevertheless, the source of the $H_2$ is yet to be determined. It was initially suggested that water impurity may be the source of $H_2$ (ref. [33]). At the anode side, $H_2O$ can be reduced to hydroxide and $H_2$ according to $H_2O + e^- \rightarrow OH^- + \frac{1}{2} H_2$. However, by quantitative analysis, Wu et al. found that residual moisture is not enough to produce the detected $H_2$, because rather ~2100 p.p.m. $H_2O$ is needed[34]. Then Gasteiger et al. suspected that an electrolyte oxidation species, such as the R–H$^+$, might be the source of $H_2$ (ref. [16]). Conversely, by isotope-labeling, Hashimoto et al. proved that the $H_2$ is not derived from the electrolyte, but rather that H impurities come from the cathode[35]. Nevertheless, recently Cui et al. reported $H_2$

evolution from the graphite anode with lithium dendrite formation, suggesting the $H_2$ may come from the PVDF/SBR binder[36,37]. Furthermore, the $H_2$ gas generation is even more critical under abusive conditions, such as battery thermal failure. Baird et al. summarized the gas composition analysis from the literature over two decades and found that $H_2$ can account for up to 40% of battery vent gases[32]. However, their source is unclear. Therefore, a comprehensive understanding of gas incubation and evolution is needed.

To provide more insights into the aforementioned issues, we in this work carried out in situ synchrotron high-energy XRD characterization coupled with mass spectroscopy (MS) at the Advanced Photon Source during heating of a lithiated graphite anode, see the in situ experimental scheme in Fig. 1a. (Supplementary Fig. 1 for a lithiated anode sample preparation and Supplementary Fig. 2 for a detailed experimental description). The phase evolution in the lithiated graphite during heating could be traced by distinct XRD reflections, as indicated by time-resolved high-energy XRD data (Supplementary Fig. 3). Meanwhile, the gas evolution due to SEI decomposition and consequential reactions could be tracked in situ and quantitatively analyzed by MS (Supplementary Fig. 2c). Our results showed that, during heating from room temperature to 280 °C, the polyethylene oxide (PEO) oligomer in the SEI decomposed first, starting between 40 and 60 °C, then a typical deintercalation staging effect of graphite was identified through heating, originating from the leaching of lithium. The generated gas species (such as $H_2$, $CH_4$, $CO_2$, CO, and $O_2$, etc.) were quantitively monitored during the whole process. DSC characterization with oxygen exposure experiment further proved the existence of highly reactive residual lithium, which triggered significant heat generation. In situ pair distribution function (PDF) analysis provided convincing evidence that the residual lithium on the anode surface is within a nano-cluster. Our work depicts SEI decomposition, lithium leaching, and gas evolution of the lithiated anode through the thermal degradation, clarifying the essential role of the SEI on the stability of lithiated graphite, and revealing the potential safety hazards posed by the flammable gas and residual liquid lithium.

## Results

With the experiment setup in Fig. 1a and Supplementary Fig. 2, the obtained phase evolution of the lithiated graphite anode during heating is shown in Fig. 1b (contour plots in Supplementary Fig. 4). During heating, the $d$-spacing evolution revealed the phase transformation pattern of $LiC_6 \rightarrow LiC_{12} \rightarrow LiC_{18} \rightarrow$ graphite, exhibiting a typical staging effect of graphite (see the simulated standard reflections of $LiC_6$, $LiC_{12}$, $LiC_{18}$, and graphite in Supplementary Fig. 3). This process is very similar to the deintercalation of lithiated graphite[38], indicating that lithium is gradually removed from the graphite anode lattice through heating. Although a similar lithium deintercalation phenomenon on the lithiated graphite anode was recently observed by Oka et al.[26], the critical SEI decomposition process elucidated below was significantly missing. In addition, to exclude the possibility of reactions between pure graphite, the PVDF binder, and conductive additives, the in situ high-energy X-ray diffraction (HEXRD) analyses of the heated unlithiated graphite anode with the same content of PVDF binder and conductive additives (carbon black) (95:2:3) were performed. As shown in Supplementary Fig. 5, without lithium intercalation, the graphite showed no phase transformation even when heated to 280 °C. The results confirmed that the unlithiated graphite is stable against PVDF and the carbon additive even at elevated temperature, while the intercalated lithium is the reason for the instability of the lithiated graphite anode.

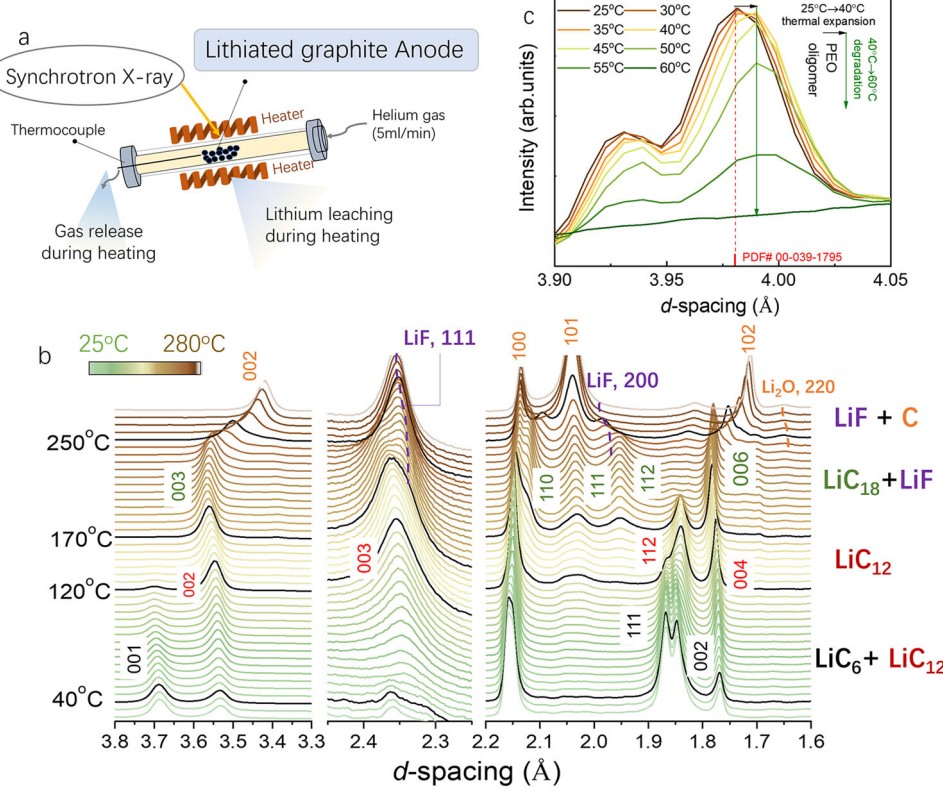

**Fig. 1 The structural evolution of lithiated graphite during the heating process. a** The scheme of the in situ experimental setup. **b** The waterfall plots of lithiated graphite anode during heating from 25 to 280 °C with 2 °C/min. The $2\theta$ value has been converted into $d$-spacing following Bragg's rule for a better comparison to the results using different X-ray sources. **c** The polyethylene oxide (PEO) oligomer lattice expansion between 25 and 40 °C, with a coefficient of thermal expansion (CTE) of $167 \times 10^{-6}$ °C, then PEO degradation during 40–60 °C.

It is widely accepted that during the initial heating the metastable part of the SEI will decompose first[1,3]. Given the importance of the SEI on battery performance, there have been numerous studies aiming to understand the chemical composition and properties of the SEI[39–43]. However, challenges still exist due to its complexity and sensitive nature, especially with regard to its thermal stability. As a result, only a few reports proposed the thermal impact on the SEI[44–46], and were mostly based on ex situ characterizations. Here, by using advanced synchrotron XRD for high phase sensitivity, we for the first time directly observed the thermal expansion and breakdown of the PEO oligomer. PEO oligomer ($-CH_2-CH_2-O)_n$ was reported as one of the major ingredients of the outer part of the SEI by XPS[47,48], NMR[49], FTIR[50], and DFT calculation[51], usually derived from ethene carbonate (EC) and ethyl methyl carbonate decomposition during the charging process of LIBs. As shown in Fig. 1c, from the initial heating starting at 25–40 °C, the PEO XRD reflection sensitively right shifted, indicating a lattice expansion (see also Supplementary Fig. 6a). The thermal expansion co-efficiency derived from the reflection at 3.98 Å is $167 \times 10^{-6}$ °C, which is well consistent with the range for polymer materials ($20–200 \times 10^{-6}$ °C), while much larger than that for inorganics, such as graphite ($8 \times 10^{-6}$ °C)[52]. Meanwhile, at this stage, the reflection intensities of LiC$_6$ (001) and LiC$_{12}$ (002) stay constant (see Supplementary Fig. 7 for Bragg reflection fitting results and Supplementary Fig. 8 for the original plots, confirming no lithium leaching at this stage). With a further temperature increase >40 °C, the PEO reflection faded and completely vanished at 60 °C, implying the breakdown of this SEI component, as shown in Fig. 1c and Supplementary Fig. 6b. Concurrently, starting at 40 °C, the LiC$_6$ Bragg reflection began to decrease with the LiC$_{12}$ increasing, corresponding to the lithium leaching out process.

With further heating, the lithium continued the deintercalation until at ~120 °C, when the LiC$_6$ phase faded; then, at ~170 °C, the LiC$_{12}$ phase vanished; finally, starting at ~250 °C, the LiC$_{18}$ phase also disappeared with the onset of the graphite 2H phase, shown in Fig. 1b.

Furthermore, the Rietveld refinement of the HEXRD patterns were performed to quantitively reveal the phase ratio of the graphite anode before and after being heated to 280 °C. As shown in Fig. 2a, the ratio of LiC$_6$ and LiC$_{12}$ in the lithiated graphite anode before heating is 49.2:50.8, indicating a designed N:P ratio (negative capacity to positive capacity ratio) of a commercial lithium-ion cell to prevent lithium dendrites, typically ~1.1:1. However, after been heated to 280 °C, the Li$_2$O (30.4 wt.%) and LiF (5.1 wt.%) phases also exist in addition to the majority of graphite 2H phase (64.6 wt.%); see Fig. 2b, indicating that the extracted lithium was converted to Li$_2$O and LiF during heating. The enlarged Bragg reflection plots focused on tracing the LiF (111, 200) and Li$_2$O (111, 220) reflections during heating are shown in Fig. 2c, d, respectively. As shown, the formation of LiF and Li$_2$O started at ~170 and 180 °C, respectively.

The reaction between the lithiated anode and PVDF/SBR binders, which accounts for a major heat generation reaction on the anode side, is usually reported by calorimetry characterizations, such as DSC and accelerated rate calorimeter (ARC)[18,53,54]. However, more than one exothermic reaction exists at high temperature (>150 °C)[53]. Therefore, only monitoring the heat is not convincing enough to ascertain the reaction. Here, for the first time, we certified this reaction by tracing the reaction products, which are LiF and H$_2$ gas. Starting at ~170 °C, the leached lithium can react with PVDF binder and lead to the formation of

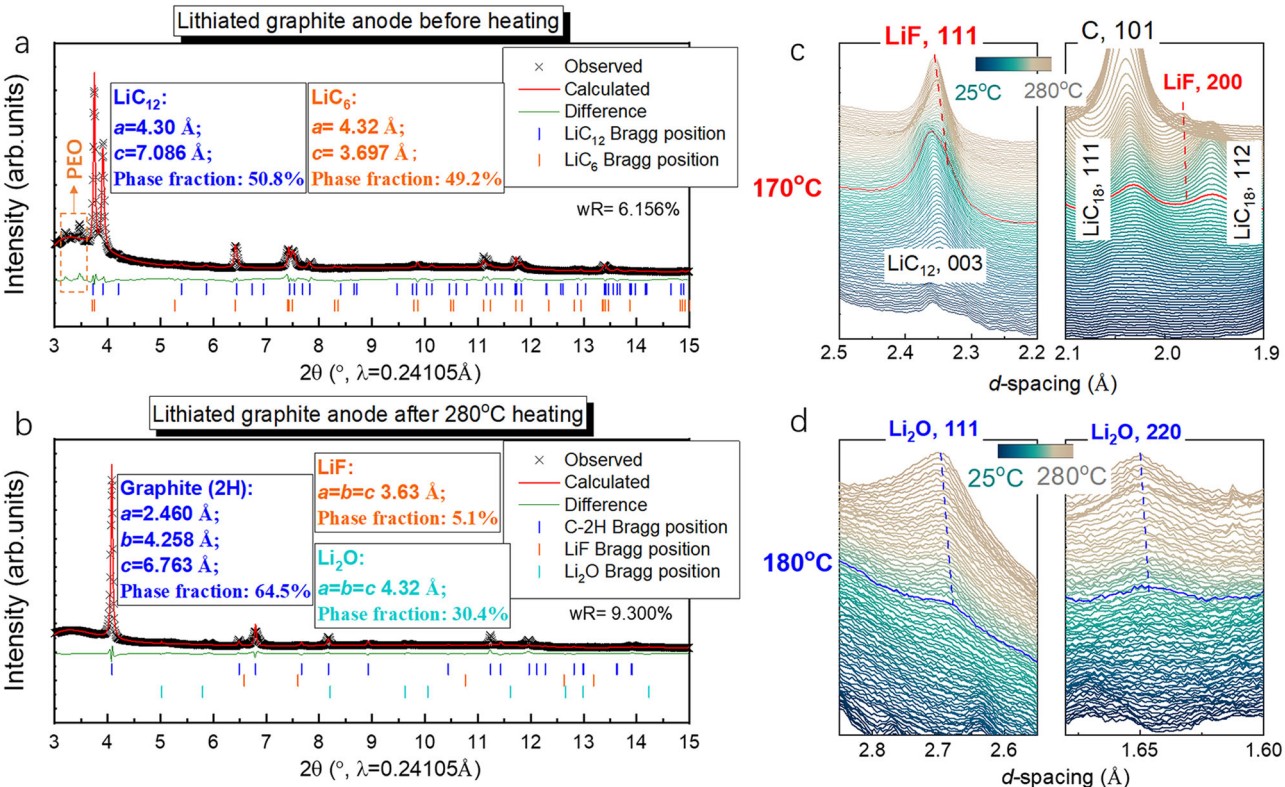

**Fig. 2 The LiF and Li₂O formation during the heating process. a** The Rietveld refinement of lithiated anode before heating. **b** The Rietveld refinement of lithiated anode after heating to 280 °C. **c** The formation of LiF by tracing the LiF (111) and (200) Bragg reflections. **d** Li₂O formation by tracing the Li₂O (111) and (220) Bragg reflections.

LiF and H₂, as shown in reaction (1):

$$Li + (-CF_2 - CH_2-)_2 \rightarrow H_2(g) + LiF$$
$$+ (-CH = CH - CH = CH-) (\sim 170\,°C) \text{ (refs.}^{18,36,53,54}) \quad (1)$$

Tracing the gas evolution can significantly enhance our understanding of the battery chemistry, for example, the SEI formation and cathode degradation[4,5,29,55,56]. This is increasingly important in battery failure analysis, because of the release of flammable and toxic gases at high temperatures, such as H₂, CH₄, and C₂H₄, incontrovertibly raises the explosive hazard[30,57,58].

Here, multiple gas species H₂ ($m/z = 2$), CH₃• ($m/z = 15$), CH₄ ($m/z = 16$), C₂H₄/CO ($m/z = 28$), O₂ ($m/z = 32$), CO₂ ($m/z = 44$), C ($m/z = 12$), CH• ($m/z = 13$), and NO₂ ($m/z = 46$) were traced and quantitatively analyzed during the heating process of a lithiated graphite anode, as shown in Fig. 3a–f and Supplementary Figs. 9–11. All the generated gas intensity is normalized according to the intensity of the helium ($m/z = 4$) flow gas set at 5 mL/min. The H₂, CH₄, CO/C₂H₄, and CO₂ gases were four major gas species during the initial heating process from room temperature to 170 °C. Although the main signal of CO at $m/z = 28$ is overlapped signal from C₂H₄, they can be distinguished by means of specific fragmentation patterns, as CO has a fragmentation pattern of carbon at $m/z = 12$ (Supplementary Fig. 9), while C₂H₄ usually does not[59]. We can then assign the first peak of $m/z = 28$ to CO and the second peak starting after 170 °C to C₂H₄, since there is no C fragment during this stage.

Enlarged gas signals focusing between 40 and 80 °C are shown in Supplementary Fig. 12, in which the CH₄, CO/C₂H₄, and CO₂ channels have shown a sharp increase between 45 and 65 °C, corresponding to the SEI decomposition between 40 and ~60 °C as observed by HEXRD. Due to the tubing used for gas transition, the gas signal shows a slight delay (5 °C). The PEO oligomer

$(-CH_2-CH_2-O-)_n$ degrades predominantly by random chain scission of the backbone with the elimination of smaller fragments, such as CO/C₂H₄, CH₄, and CO₂ (reaction (2))[60,61]. The decompositions of other SEI components also possibly lead to releases of $m/z = 16$, 28, and 44 gases, such as lithium ethylene dicarbonate (LEDC), Li₂CO₃, ROCO₂Li, ROLi (R= CH₃–, CH₃–CH₂–, CH₃–CH₂–CH₂–, etc.), following the below reactions (reactions (3–5))[15,55,62,63]:

$$(-CH_2 - CH_2 - O-)_n \rightarrow H_2(g) + CH_x + CO(g)$$
$$+ H_2O + (-CH_2 - O-)_{n-2} \text{ (refs. }^{60,61}) \quad (2)$$

$$LEDC \rightarrow Li_2CO_3 + C_2H_4(g) + CO(g) + CH_3CH_2CO_2Li \text{ (refs. }^{15,55,62}) \quad (3)$$

$$ROCO_2Li + H_2O \rightarrow Li_2CO_3 + CO_2(g) + LiOR \text{ (refs. }^{15,55,62,63}) \quad (4)$$

$$(CH_2OCO_2Li)_2 \rightarrow Li_2CO_3 + C_2H_4(g) + CO_2(g) + O_2(g) \text{ (refs. }^{55,63}) \quad (5)$$

At ~100 °C, the maximum gas release rate of the CH₄, CO/C₂H₄, O₂, and CO₂ was 0.06, 0.09, 0.02, and 0.25 μL/min/mg, respectively, with a cumulative ratio of 7.3% (CH₄), 21.2% (CO/C₂H₄), 10.1% (O₂), and 22.0% (CO₂); see Fig. 3g and Supplementary Fig. 13. The ratio between CO/CO₂ and CH₄ indicated that the decomposition of lithium alkyl carbonates (following reactions (3–5)) is the main gas release reaction, compared with PEO oligomer degradation (with CH₄ production, reaction (2)).

As for the H₂ gas, see Supplementary Fig. 12, the H₂ intensity increased exponentially as the temperature increased from 40 to 80 °C, in contrast with the appearance of a plateau at ~70 °C in

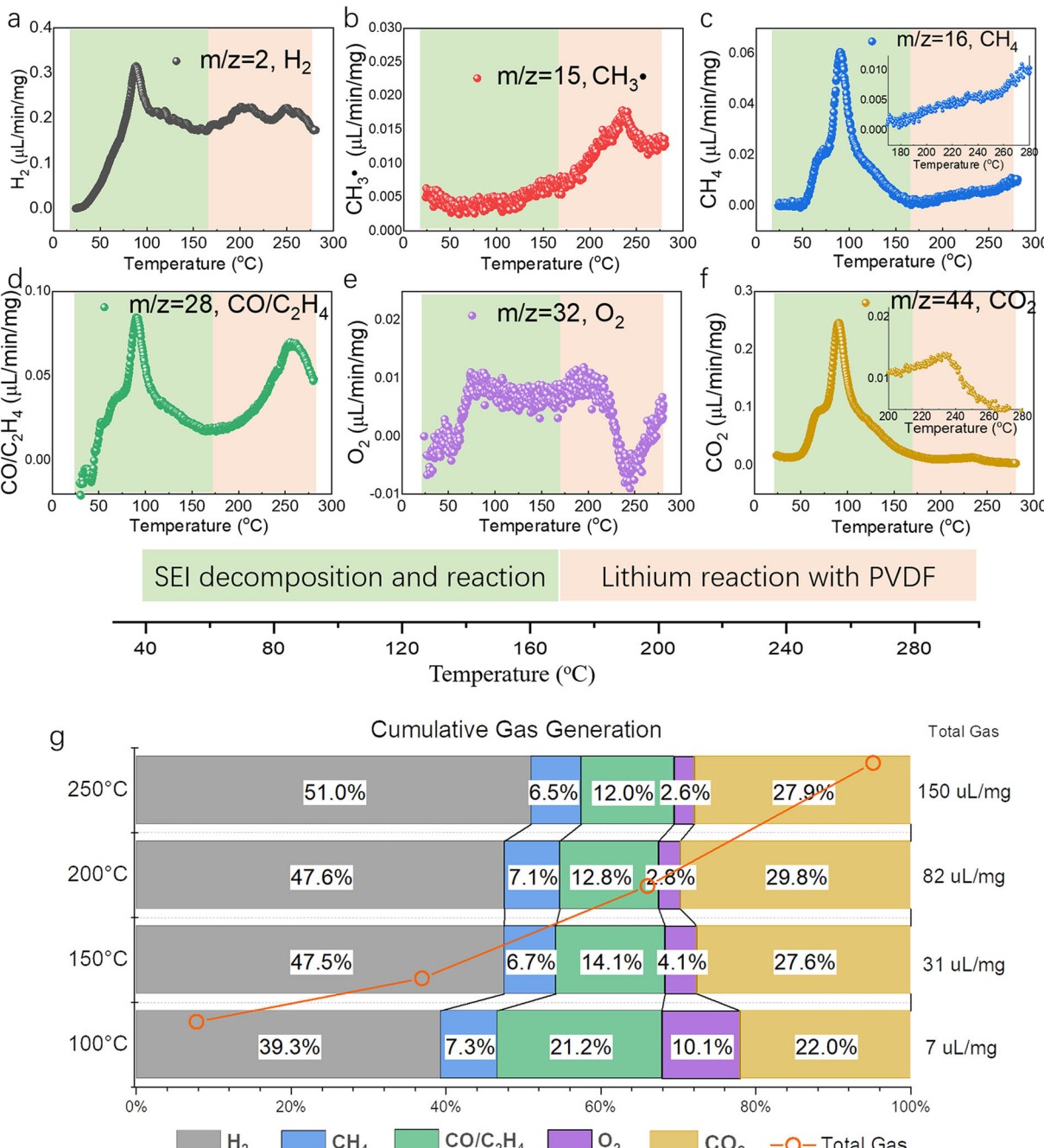

**Fig. 3 The in situ gas release of lithiated graphite anode during heating. a–f** The in situ monitoring of $H_2$, $CH_3$, $CH_4$, $CO/C_2H_4$, $O_2$, and $CO_2$ gas evolution during the heating of the lithiated graphite anode. **g** The quantitative analysis of the cumulative gas ratio at 100, 150, 200, and 250 °C.

other SEI degradation-derived gas evolution curves ($CH_4$, $CO/C_2H_4$, and $CO_2$), indicating the $H_2$ was related to the leached lithium rather than the SEI decomposition. This is because the leached lithium linearly increased as revealed by HEXRD, compared to the breakdown and then disappearance of SEI. The leached lithium may also react with residual $H_2O$ or ROH for $H_2$ gas release[16,64–66]:

$$2ROH + Li \rightarrow H_2(g) + 2ROLi \text{ (refs. }^{16,64}) \quad (6)$$

$$Li + H_2O \rightarrow H_2(g) + LiOH \text{ (ref. }^{16}) \quad (7)$$

$$Li + O_2 \rightarrow Li_2O \text{ (ref. }^{65}) \quad (8)$$

The gas release intensity of $CO/C_2H_4$, $CH_4$, and $CO_2$ rapidly decreased after the majority of SEI decomposed at 100 °C; see Fig. 3c, d, f. However, the $H_2$ remains at a relatively high level of ~0.2 μL/min/mg, confirming the continuous drain out of lithium from the graphite. After 170 °C, the reaction between leached Li and PVDF binder (reaction (1)) activated. Thus, the $H_2$ intensity increased again; see Fig. 3a. Meanwhile, the gas of $CH_3$ ($m/z = 15$) fragment, $CH_4$ ($m/z = 16$), and $C_2H_4$ ($m/z = 28$) were also increasingly detected as a result of the PVDF ($–CF_2–CH_2–)_n$ decomposition reaction to eliminate the small hydrocarbon fragments. To sum up, with in situ mass spectrometry, the gas release of the lithiated anode was quantitively analyzed through heating. The $H_2$, $CO/C_2H_4$, $CH_4$, and $CO_2$ were the main gas products initially during the SEI decomposition and the

associated reaction, and after 170 °C, the leached lithium reacted with PVDF and other residual oxygen-containing ingredients, with the production of LiF, $H_2$ gas, and $Li_2O$. These two processes are indicated with light green and orange color in Fig. 3a–f. Consequently, the cumulative gas produced from the lithiated anode at 250 °C (a typical thermal runaway temperature of commercial LIBs[67]) is ~150 μL/mg$_{anode}$ with the ratios of 51% ($H_2$), 6.5% ($CH_4$), 12.0% ($CO/C_2H_4$), 2.6% ($O_2$), and 27.9% ($CO_2$). The total gas volume measured here (150 μL/mg$_{anode}$) is comparable to the report of the vented gas analysis of large-format LIBs by Yuan et al. with 36.5–245 μL/mg$_{battery}$[68], considering the fluctuation caused by gas generation from the electrolyte and cathode in commercial full LIBs. On the other hand, compared to the vented gas composition of full LIBs[30–32], the anode-released gas showed a higher $H_2$ concentration, suggesting that the $H_2$ gas is mainly driven from the anode rather than the cathode or electrolyte. Oppositely, the $CO_2$ ratio is lower than that in the full batteries, indicating the generation of $CO_2$ in the full cells comes from the decomposition of cathode and electrolytes. By the advantages of time-resolved measurement, we revealed that the one major SEI component-PEO oligomer, accounts for the initial $H_2$ gas release, by reaction (2); while the other SEI component-lithium alkyl carbonate ($ROCO_2Li$) accounts for the $CO_2$ generation, following reaction (4).

Post-analysis was conducted to further confirm the degradation pathways of the lithiated graphite; see details in the Supplementary Information. Before charging, as shown in Fig. 4a, it is observed that the mesoporous graphite anode with a particle size ~15 μm, and the PVDF binder and conductive carbon black are on the surface and gaps of graphite. During heating, the drained-out lithium from the graphite lattice can gather around at the edges of graphite flakes (moving out along the in-plane direction), as illustrated in Fig. 4b. The accumulated lithium can react first with the SEI components. Then, with a further temperature increase, the lithium reaction with the PVDF binder happened with LiF formation and $H_2$ gas release. Therefore, as shown in Fig. 4c, a thick layer of reaction products was observed after reaching 280 °C (anode collected after natural cooling), covering the anode surface. Moreover, the vigorous reaction of drained lithium can cause the exfoliation of the graphite layer. In the cases of a full battery with a liquid electrolyte or a cathode oxygen gas release, such exfoliation of graphite may exponentially increase the reaction area and expose more lithium through the opened graphene layer, thus leading to the rapid heat generation.

Note that, due to the limited amount of PVDF binder (anode composition 95 wt.% graphite, 2 wt.% PVDF, and 3 wt.% carbon black), only 5.2 wt.% of the total leached lithium is consumed by reaction (1); with another 55.5 wt.% of leached lithium consumed by the SEI layer, see Supplementary Notes 1 and 2, and Supplementary Table 1. Thus, 39.3 wt.% of the residual liquid lithium can exist on the graphite surface at high temperatures, leading to great potential hazards. To confirm this hypothesis, an oxygen exposure experiment was designed during the DSC measurements to simulate the thermal-induced oxygen release from the layered cathode[5,20] and/or the solid-state electrolyte[65] in a full battery; see Fig. 4d. The DSC purge gas was swift from inert nitrogen to oxygen at the designated high temperatures, such as 200, 240, and 260 °C. As shown, with oxygen exposure, an immediate heat generation peak was triggered, with 123, 795, and 852 J/g at 200, 240, and 260 °C, respectively. As indicated by HEXRD, the phase composition at 200, 240, and 260 °C can be identified to calculate the weight ratio of leached lithium; see Supplementary Table 1. At 200 °C, almost all the leached lithium (3.72 wt.%, based on the total weight of lithiated graphite) reacted with the SEI, leaving limited amount of residual leached lithium, as indicated by the heat generation of only 123 J/g. However,

from 200 to 240 °C, 2.1 wt.% more lithium was drained from the graphite anode. Based on the reaction entropy of $4Li + O_2 \rightarrow 2Li_2O$ ($\Delta H = -299.4$ kJ/mol$_{lithium}$)[65], theoretically, at most 905.96 J/g (2.1% × 299.4/6.94 kJ/g) heat would be generated (see Supplementary Note 2). This is within the same order of magnitude as our experimental observation of 795 J/g, indicating good consistency in battery heat generation[54]. With the temperature further increased to 260 °C, the theoretical heat generation with 2.98 wt.% (2.1 + 0.88wt.%) residual lithium was 1285.6 J/g, compared to 852 J/g with the experimental measurement. Considering that the leached lithium can also be consumed in other reactions, as indicated by gas release and $Li_2O$ formation at high temperatures, a lower measured heat generation is quite reasonable. Note that only the lithiated graphite anode evolved in this experiment, and at this temperature, the reaction between graphite and oxygen was not activated (graphite + $O_2 \rightarrow CO_2$ occurred at 400–1200 °C, ref. [69]), which is also indicated by the limited amount of $CO_2$ generation at high temperature (Fig. 3f). The existence of residual lithium is the most possible answer that accounts for this large heat generation. Traditionally, the flammable carbonate electrolyte was considered as a "fuel tank" due to a large energy release with combustion through thermal runaway. Here, for the first time, we revealed that the residual lithium leached out on the graphite anode surface is the true source accounting for this significant energy release during thermal failure.

Furthermore, although the process of lithium draining was confirmed by HEXRD phase evolution ($LiC_6 \rightarrow LiC_{12} \rightarrow LiC_{18} \rightarrow$ graphite) and the residual lithium was indicated by DSC heat generation, there was no direct ordered lithium body-center-cubic (bcc) structure identified through the heating by in situ HEXRD, as shown in Supplementary Fig. 14. If the well-ordered bcc stacking of lithium atoms was formed with a long enough coherence length (Fig. 4e), the reflection at 5.56°, indexing the most intense scattering of Li (110) plane, should have been observed[70]. However, there is no such Bragg reflection throughout the heating. To probe the structure of the leached lithium, we further conducted in situ PDF analysis (see experimental details in Supplementary Fig. 15), which is a useful method for analyzing both crystalline and amorphous materials[71]. As shown in Fig. 4f, the calculated PDF patterns of $LiC_6$, $LiC_{12}$, $LiC_{18}$, and bcc-stacked lithium are also present in the figure for a better peak index. The primary PDF peaks at 1.41, 2.45, 2.83, and 3.75 Å are attributed to the well-ordered C–C bonding of the in-plane graphene layer[72] (see Supplementary Fig. 16 for illustration). C–C bonding, especially those with a short range (1.41, 2.45, and 2.83 Å), remained unchanged during the heating, indicating a stable in-plane carbon host structure. As for the Li–Li bonding, a shoulder peak at 3.51 Å was identified at high temperatures of 280 and 320 °C. However, compared with the pure liquid lithium PDF revealed by Chen and Salmon et al.[73,74], the longer-range Li–Li peak (5.82 Å) was not observed in our case. The result may indicate that the leached lithium from graphite was well-distributed and formed nano-cluster aggregates, thus exhibiting no long-range order after the first coordination sphere (3.5 Å). In addition, with nano-cluster lithium, the high surface area greatly boosted its reactivity, leading to more severe safety concerns.

## Discussion
By using in situ synchrotron XRD coupled with MS and X-ray scattering PDF analysis, the unprecedented picture of battery thermal failure on the anode side was comprehensively illustrated here. As shown in Fig. 5a, a stable SEI layer consisting of LEDC, PEO oligomer, $Li_2CO_3$, LiF, and lithium alkyl carbonates, etc., formed on the graphite anode after the formation process. During

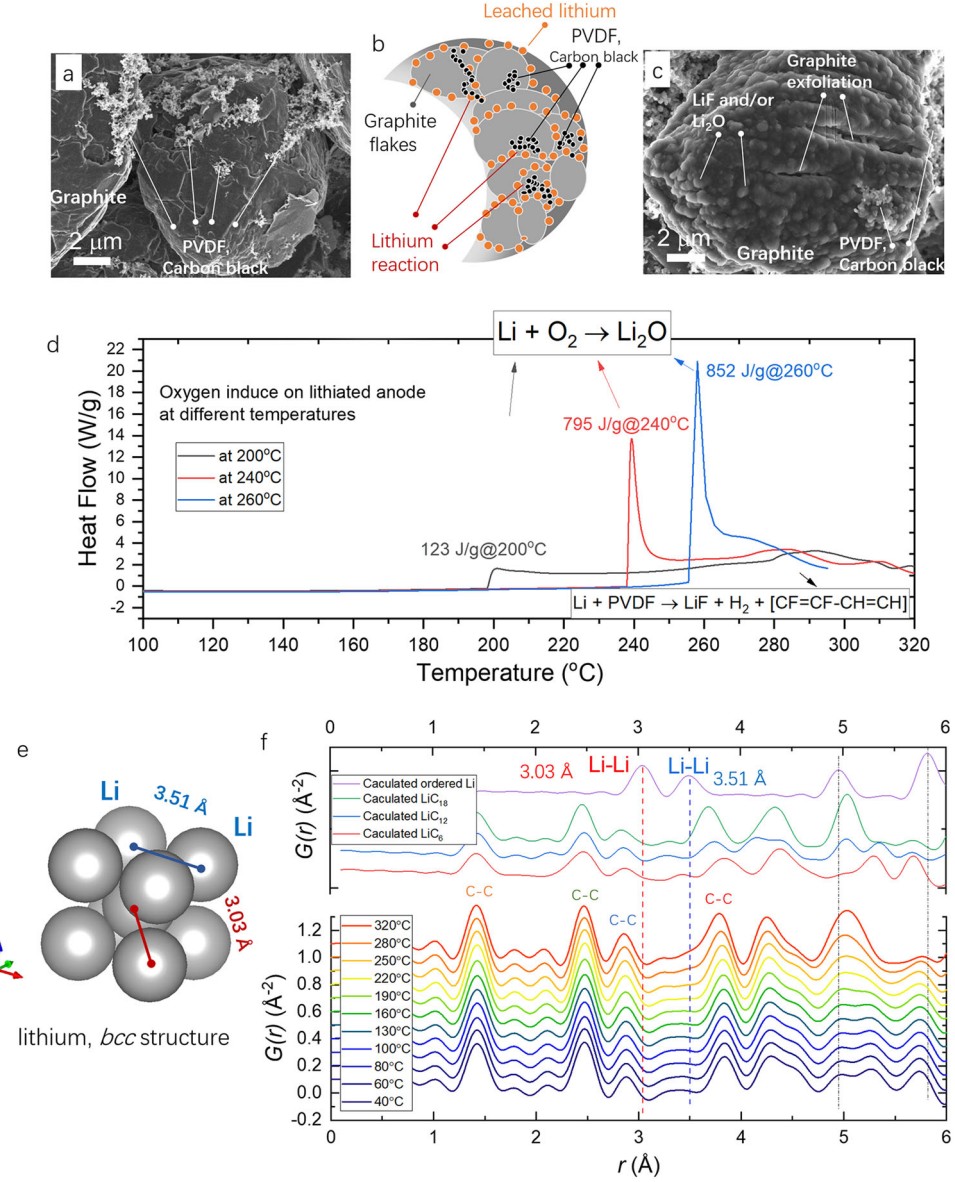

**Fig. 4 The post-analysis and in situ PDF of the lithiated graphite anode. a** SEM image of the graphite anode before heating. **b** An illustration of the lithium leaching and reaction process on the edges of graphite flakes. **c** SEM image of the lithiated graphite anode after 280 °C heating. **d** The influence of oxygen on lithiated graphite anode at different temperatures during the DSC measurements. **e** The illustration of lithium bcc atomic structure. **f** The in situ PDF analysis of the lithiated graphite anode during the heating.

the initial heating, see Fig. 5b, the PEO oligomer starts to decompose from 40 °C, followed by the degradation of LEDC and $ROCO_2Li$ until ~100 °C. The lithium begins to leach out after the SEI damage (denoted by the blue arrow) and reacts with the remaining SEI ingredients, thus leading to increased $H_2$, $O_2$, $CH_4$, and $CO_2$ gas releases. The cumulative gas is 7 μL/mg$_{anode}$ with ratios of 39.3% ($H_2$), 7.3% ($CH_4$), 21.2% ($CO/C_2H_4$), 10.1% ($O_2$), and 22.0% ($CO_2$) at 100 °C. Then starting at 170 °C, the reaction between lithium and PVDF leads to the formation of LiF and more $H_2$ gas; Fig. 5c. Nevertheless, not all the leached lithium is consumed by the SEI and PVDF. With a further temperature increase to 180 °C, the residual lithium starts to melt to further increase its reactivity with a lower density, higher porosity, and tortuosity at the liquid amorphous state[71]. Therefore, 39.3 wt.% of residual liquid lithium existing on the graphite surface raises a major safety concern, see Fig. 5d, because it is ready to react with cathode/SSE-released oxygen and/or the liquid electrolyte to trigger thermal runaway. In addition, at 250 °C, the large portion

of flammable gas also raises a great explosive hazard, as the cumulative gas is 150 μL/mg$_{anode}$ with 51% ($H_2$), 6.5% ($CH_4$), and 12.0% ($CO/C_2H_4$).

In practical LIBs, the electrolyte exists as one important component that affects battery safety. Thus, the in situ HEXRD analysis of the degradation of the lithiated graphite degradation with the presence of electrolyte was conducted further; see the results in Fig. 6 and the contour plots in Supplementary Fig. 17. As shown, with the presence of electrolyte, the lithiated graphite anode follows similar phase transformation pathways as the dry lithiated graphite, exhibiting the typical $LiC_6 \rightarrow LiC_{12} \rightarrow LiC_{18} \rightarrow$ graphite phase evolution. However, the reaction kinetics were increased significantly. Specifically, the $LiC_6$ (001) reflection started to decrease immediately upon heating (Fig. 6a), compared to 60 °C in the case of dry lithiated graphite. Moreover, with the presence of electrolyte, the LiF (110) reflection started to appear at ~114 °C, compared to 170 °C for the dry lithiated graphite, suggesting the leached lithium reacted with the electrolyte; see

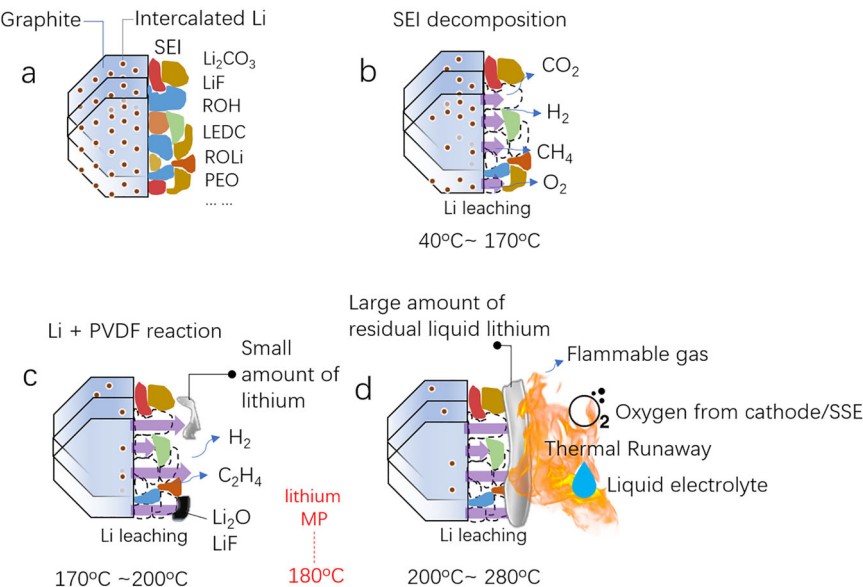

**Fig. 5 The thermal degradation pathway of lithiated graphite anode and its safety hazards. a–c** The thermal evolution of the lithiated graphite anode revealed by multiple in situ characterizations. **d** The safety hazards posed by flammable gas and residual nano-cluster liquid lithium of the lithiated anode at high temperatures, which are ready to react with the cathode (or solid-state electrolyte, SSE)-released oxygen and/or liquid electrolyte to trigger thermal runaway; the lithium melting point of 180 °C is marked.

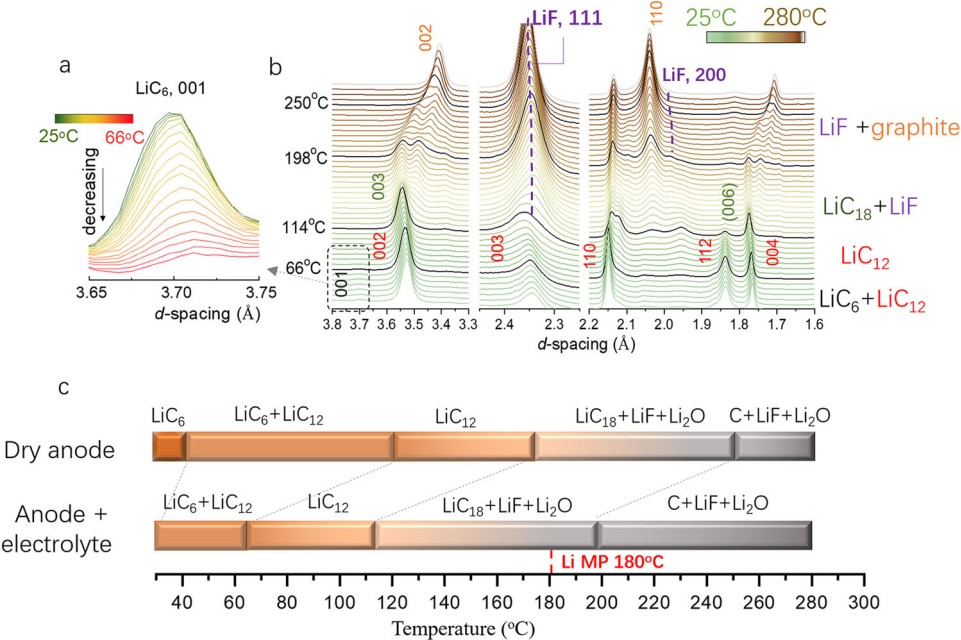

**Fig. 6 The structural evolution of lithiated graphite with the electrolyte during heating. a** The HEXRD plots of $LiC_6$ (001) Bragg reflection evolution from 25 to 66 °C. **b** The HEXRD plots of lithiated graphite anode with electrolyte from 25 to ~280 °C with 2 °C/min. The $2\theta$ value has been converted into d-spacing following Bragg's rule. **c** The comparison of the phase evolution of the lithiated graphite anode with and without the presence of electrolyte. The lithium melting point (MP) 180 °C is also described in the figure.

Supplementary Fig. 18. The disappearance of $LiC_{18}$ (003) started at 198 °C in comparison with 250 °C for the dry lithiated graphite anode. With that, we can summarize the phase evolution pathway of the lithiated graphite anode during the heating process with and without electrolyte, as in Fig. 6c. All phase transformations occurred at a lower temperature with an ~60 °C decrease compared to the dry lithiated graphite, implying a faster lithium leaching kinetics by reacting with the electrolytes. This is well consistent with the literature reports by DSC heat generation measurements, in which the heat generation onset temperature of the lithiated graphite decreased with the presence of electrolyte (from ~220 °C with the dry lithiated graphite to 150 °C with the electrolyte[20,53]). The accelerated decomposition of lithiated graphite with the presence of the electrolytes is because the large amount of electrolyte salt (e.g., $LiPF_6$) that can be decomposed into highly reactive products, such as $PF_5$(gas), $OPF_3$, and HF during heating[75], and electrolyte solvents (e.g., EC) that can be reduced at high temperatures[76], both of which can react simultaneously with the metastable decomposition products in the SEI layer or leached lithium, therefore, lower the onset reaction

temperature. In addition, the hidden formation of highly reactive lithium revealed here can explain the reaction mechanisms that are behind various thermal characterizations, such as DSC and ARC. For example, the thermal runaway of LIB with non-flammable electrolyte revealed by Hou et al.[21] might not be triggered by the reactions between nonflammable electrolyte and $LiC_6$, but more accurately between electrolytes and the residual liquid lithium on graphite surface; and the thermal runaway of LIBs without internal short circuit presented by Liu et al.[20], should be actually attributed to the reaction between delithiated-cathode-released oxygen and liquid lithium at 231 °C, rather than the $LiC_6$. To effectively mitigate their reactions, one might focus on rational surface coating on graphite to prevent their physical contact. As a result, our present findings will stimulate more rational designs for safe LIBs.

Furthermore, it is interesting to notice the remarkable disparity in the thermal stability between the chemically prepared $LiC_6$ (refs. [17,22,23]) and electrochemically activated $LiC_6$ from batteries. The major difference between those two $LiC_6$ lies in the SEI surface layer. Without the SEI, the chemically processed $LiC_6$ is thermodynamically stable even up to 350 °C (refs. [17,22,23]). In contrast, the $LiC_6$ with the SEI layer decomposes immediately after the SEI breakdown starting at 40 °C. This, on one hand, calls for more attention to constructing a robust SEI to preserve the lithium in the graphite anode. On the other hand, it shows a bright future for our battery community, as the safety concerns about $LiC_6$ can be thermodynamically alleviated like chemically prepared $LiC_6$. Strategies for tuning the lithium reaction kinetics, such as surface coating or doping, should be taken into consideration.

In summary, with multiple in situ synchrotron X-ray characterizations, the phase evolution pattern, gas release pathway, and safety hazards of residual liquid nano-cluster lithium during the thermal degradation of the lithiated graphite anode were comprehensively illustrated. The above understanding and underlying mechanism first emphasized the role of the SEI in anode thermal stability protection. In order to prevent the lithium leaching and further reaction, a robust SEI is urgently needed. The second highlight uncovered here is the source of the flammable gas, such as $H_2$. We showed that the initial $H_2$ release is triggered by the SEI decomposition and the Li reaction with SEI-rated ingredients; then, after 170 °C, the reaction between lithium and PVDF produced more $H_2$. A binder selection should be carefully evaluated in the future to reduce the flammable gas hazard. Last but not least, the first-time identified residual lithium on the graphite anode surface is of great importance for battery safety. Further studies on kinetically tuning of its nucleation process and mitigation technologies, such as surface coating and structural design, are highly recommended in order to control its reactivity. Our findings will stimulate extensive efforts to enable safe energy storage systems with rational design.

## Methods

The lithiated graphite anode was collected from a commercially available 24 Ah vehicle-use pouch LIB. The cathode active material is $LiNi_{1/3}Mn_{1/3}Co_{1/3}O_2$, the anode composition is graphite (95 wt.%), PVDF (2 wt.%), and carbon black (3 wt.%), the separator is polyethylene. The battery was charged to 4.2 V under constant current (1/3 C)-constant voltage (4.2 V, 1/20 C cutoff current) after being charged for two cycles, then the pouch cell was carefully disassembled inside a glovebox. The lithiated graphite electrode was rinsed with dimethyl carbonate solvent followed by drying under vacuum overnight, then the lithiated graphite powder was carefully scratched from the electrode and ready for characterizations. More details about battery specifications and cycling performance are shown in the Supplementary Information.

Supplemental characterization procedures, such as in situ high-energy XRD during heating with mass spectrometry, DSC test with oxygen induction, SEM post-analysis, and in situ PDF measurement and analysis, are detailed in the Supplementary Information.

## Data availability

The data that support the findings of this study are available from the corresponding authors G.-L.X., M.O., and K.A. upon reasonable request.

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

## Acknowledgements

Research at Argonne National Laboratory was funded by the US Department of Energy (DOE) Vehicle Technologies Office. Support from Tien Duong of DOE's Vehicle Technologies Office is gratefully acknowledged. Use of the Advanced Photon Source was supported by the US Department of Energy, Office of Science and Office of Basic Energy Sciences, under Contract No. DEAC02-06CH11357. The Ministry of Science and Technology of China is acknowledged under Grant No. 2019YFE0100200. X.L., G.-L.X., M.O., and K.A. acknowledge the support of the U.S. China Clean Energy Research Center (CERC-CVC2).

## Author contributions

X.L. and G.-L.X. conceived the idea; X.L. performed in situ heating HEXRD with gas analysis and processed the data; L.Y. and X.L. carried out the in situ heating PDF and analyzed the data; D.R. prepared the lithiated graphite anode and conducted the DSC measurement; L.W., Y.R., W.X., S.L., H.W., X.H., and Z.C. helped with data analysis. X.L. and G.-L.X. prepared the manuscript; G.-L.X., M.O., and K.A. managed the project; and all authors contributed to discussions and paper revisions.

## Competing interests

The authors declare no competing interests.
