## [Peer Review File · Nature Communications]

Reviewer #1 (Remarks to the Author):

The authors reported the thermal impact on solid-electrolyte interphase (SEI) decomposition, lithium leaching, and gas release of the lithiated graphite, which is detected by synchrotron X-ray in combination with mass spectroscopy. The authors found that the SEI decomposition, lithium leaching, and SEI residues-lithium reaction are dominant at low temperatures (40-170°C), while at higher temperatures (>170°C) the reaction of leached lithium with PVDF binder prevails. The SEI decomposition can generate the flammable H₂ gas. Another source for generating H₂ is the interaction between leached lithium with PVDF. Because the limited amount of PVDF and SEI component, a large amount of residual liquid lithium are present at higher temperatures, which is dangerous to cause thermal runaway. The presented results are convincing, novel and informative. The reviewer recommends accepting the manuscript in Nature Communications after addressing some issues.

General remarks

1. The authors discussed mainly the results on the dry lithiated anode without the presence of liquid electrolyte. From the reality perspectives, the results with liquid electrolyte are more convincing and deserve more discussions, although they are delivering similar information. So why not discuss the results with liquid electrolyte by more wording..?
2. During heating the dry lithiated anode, Lithium will be leached by the thermal driving force. If considering the full battery is being heated in the occasion that the battery is simultaneously under discharging (lithiated anode will be delithiated), will the lithium will remain liquid lithium at anode surface or be intercalated back to cathode...? Could the authors comment on this..?

Specific comments:

1. At around 170°C, almost half amount of lithium will be leached out from the lithiated anode, since LiC₆ will be converted into LiC₁₂. In principle, the leached lithium will readily react with the O₂ gas, which may lead to sharp decrease of this gas in the atmosphere. Why it is not revealed in Figure 3e...?
2. On page 12, the author reasoned the residual Lithium is a potential hazard to the battery safety. Does it mean, for the batteries' storage or transportation process, it would be much safer to maintain the battery in fully discharged state? So lithium should better stay in cathode side to avoid anode leaching in the case of any thermal abuse..?
3. Also on page 12, it is stated "only 5.2 wt.% of the total leached lithium is consumed by reaction (1)". This is a bit misleading because leached lithium can also react with SEI component, gas... Shouldn't the total amount of lithium (leached) consumption by all factors be considered/calculated...?
4. Should the authors add "O₂" in Figure 5b, which is coming from equation 5...?

Chunguang Chen

Reviewer #2 (Remarks to the Author):

The authors reported the work entitled "In situ observation of thermal-driven SEI decomposition, gas evolution, and lithium leaching in lithiated graphite anode". Solid-electrolyte interphase (SEI) decomposition, lithium leaching, and gas release of the lithiated graphite anode during heating were

investigated by in situ synchrotron X-ray techniques and in situ mass spectroscopy. This work discovered the critical role of SEI in anode thermal stability and the potential safety concerns of flammable gases and leached Li. However, some issues need to be resolved before its publication in Nature Communications.

1. The authors demonstrate the direct Li leaching from lithiated graphite upon heating. However, it is still unclear why and how Li leaching happens from lithiated graphite upon heating. The underlying mechanism should be clarified by combining DFT or literature.
2. It is written "charged graphite anode or charged anode" in Fig. S2a and 2b, which can confuse since the "Charged graphite anode or charged anode" can be different in full cell and half cell. It is better to use "lithiated graphite anode" to avoid confusion as always used in the whole manuscript.
3. The electrolyte composition of the pouch cell was missing in the materials part (Fig. S1c and text). The full name of EC and EMC is missing in SI. It is written, "PEO oligomer (-CH₂-CH₂-O)_n was reported as one of the major ingredients of the outer part of the SEI by XPS, NMR, FTIR, and DFT calculation, usually derived from ethene carbonate (EC) decomposition during the charging process of LIBs." If it contains EMC or Dimethyl carbonate (DMC) in the electrolyte, why the authors did not correlate the EMC or DMC-derivation containing SEI with gas formation in the manuscript?
4. References are missing for PDFgui software in SI and how the PDF data in Fig.4f was obtained should be briefly described.
5. According to the International Union of Crystallography (IUCr), the correct form of indices of Bragg reflection in diffraction patterns should be without bracket. In diffraction patterns, one should call "reflection", not "peak". Please correct all the related parts throughout the whole manuscript.
6. There are some typos such as "degredation→ degradation" in Fig. 1d, "fiction→ fraction" (Fig.2).
7. Did the authors perform in situ HEXRD and PDF for standard materials such as NaCl or LaB₆ to calibrate the temperature for each pattern, which is very important to reach each "target or marked" temperature for each diffraction or PDF pattern?
8. How did the authors determine the LiF and Li₂O separately in SEM (Fig.4c)?
9. The CO₂ intensity almost has the same order as H₂ as shown in Fig.3, what this indicates, and if it can also be correlated to something?

Reviewers' Comments:

Reviewer #1 (Remarks to the Author):

*The authors reported the thermal impact on solid-electrolyte interphase (SEI) decomposition, lithium leaching, and gas release of the lithiated graphite, which is detected by synchrotron X-ray in combination with mass spectroscopy. The authors found that the SEI decomposition, lithium leaching, and SEI residues-lithium reaction are dominant at low temperatures (40-170°C), while at higher temperatures (>170°C) the reaction of leached lithium with PVDF binder prevails. The SEI decomposition can generate the flammable H₂ gas. Another source for generating H₂ is the interaction between leached lithium with PVDF. Because the limited amount of PVDF and SEI component, a large amount of residual liquid lithium are present at higher temperatures, which is dangerous to cause thermal runaway. The presented results are **convincing, novel and informative**. The reviewer recommends accepting the manuscript in Nature Communications after addressing some issues.*

General response: We thank the reviewer for the recommendation and valuable suggestion. The detailed point-by-point responses are attached below:

General remarks

1. *The authors discussed mainly the results on the dry lithiated anode without the presence of liquid electrolyte. From the reality perspectives, the results with liquid electrolyte are more convincing and deserve more discussions, although they are delivering similar information. So why not discuss the results with liquid electrolyte by more wording..?*

Response: Thanks for the suggestion. We agree with the reviewer that the lithiated graphite anode with the presence of electrolytes is more comparable to real scenarios. We have added additional discussion in the page 19 of revised manuscript as following:

‘The accelerated decomposition of lithiated graphite with the presence of the electrolytes is because the large amount of electrolyte salt (e.g. LiPF₆) that can be decomposed into highly reactive products such as PF₅(gas), OPF₃ and HF during heating [Thermochimica Acta 480, 10-14], and electrolyte solvents (e.g. EC) that can be reduced at high temperatures [Energy Storage Materials, doi.org/10.1016/j.ensm.2021.04.035], both of which can react simultaneously with the metastable decomposition products in the SEI layer or leached lithium, therefore lower the on-set reaction temperature. In addition, the hidden formation of highly reactive lithium revealed here can explain the reaction mechanisms that are behind various thermal characterizations such as DSC and ARC. For example, the thermal runaway of lithium-ion battery with non-flammable electrolyte revealed by Hou *et al.*²¹ might not be triggered by the reactions between non-flammable electrolyte and LiC₆, but more

accurately between electrolytes and the residual liquid lithium on graphite surface; and the thermal runaway of lithium-ion batteries without internal short circuit presented by Liu *et al.*,²⁰ should be actually attributed to the reaction between delithiated-cathode-released oxygen and liquid lithium at 231°C, rather than the LiC₆. To effectively mitigate their reactions, one might focus on rational surface coating on graphite to prevent their physical contact. As a result, our present findings will stimulate more rational designs for safe lithium-ion batteries.'

2. *During heating the dry lithiated anode, Lithium will be leached by the thermal driving force. If considering the full battery is being heated in the occasion that the battery is simultaneously under discharging (lithiated anode will be delithiated), will the lithium will remain liquid lithium at anode surface or be intercalated back to cathode...? Could the authors comment on this..?*

Response: Thanks for this comment. Those two scenarios share the same fact that the lithium is de-intercalating from the lithiated graphite anode, while the difference is whether the lithium ions will go back into the cathode or leave on the surface of graphite. We believe that if the cell is discharging and heating simultaneously, both situations may exist, which depend on the kinetics determined by the discharge rate and heating rate. However, in general, when the cell is heated, the internal temperature of the cell will be increased dramatically to exceed the discharge rate, which will lead to the formation of residual lithium on graphite surface. Once the temperature high enough to melt lithium to liquid (as mentioned by the reviewer, >180 °C), we believe the exothermal reactions caused by liquid lithium is highly possible to cause the thermal runaway of the battery.

Specific comments:

3. *At around 170°C, almost half amount of lithium will be leached out from the lithiated anode, since LiC₆ will be converted into LiC₁₂. In principle, the leached lithium will readily react with the O₂ gas, which may lead to sharp decrease of this gas in the atmosphere. Why it is not revealed in Figure 3e...?*

Response: Indeed, as shown in Figure 3e, the intensity of O₂ gas sharp decreased starting from around 200°C. The temperature delay (~ 30°C) is because most of the leached lithium from graphite (LiC₆→LiC₁₂) **has already been consumed by the reactions with SEI and PVDF before 200°C (see Table S1 and Figure 4d)**. When the temperature reached 200°C, more residual lithium was formed on graphite surface and became liquid at 200°C, which aggravated the reactivity with O₂ and led to the observed intensity decrease of O₂ gas.

4. *On page 12, the author reasoned the residual Lithium is a potential hazard to the battery safety. Does it mean, for the batteries' storage or transportation process, it would be much safer to maintain the battery in fully discharged state? So lithium should better stay in cathode side to avoid anode leaching in the case of any thermal abuse..?*

Response: We agree with the reviewer on this point. If the lithium stays in the cathode, which means the battery is at the discharge state, which should be the most stable state for lithium-ion battery. Not only to avoid anode lithium leaching, but also to prevent the cathode from oxygen release (our previous results discussed the consequence of over-heating on the delithiated cathode, which triggered intensive phase transformation with oxygen release [*JACS*, 2020, 142, 19745]). However, in many cases, an immediate use of the battery is of high priority, for example for military serves and portable devices. Thus, the batteries have to be storage or transmitted at fully charged or half-charged state. In those cases, we believe a proper thermal managements or transportation regulations are required to prevent the battery over-heating.

5. *Also on page 12, it is stated “only 5.2 wt.% of the total leached lithium is consumed by reaction (1)”. This is a bit misleading because leached lithium can also react with SEI component, gas... Shouldn't the total amount of lithium (leached) consumption by all factors be considered/calculated...?*

Response: Thanks for pointing it out. We have revised the description accordingly. By taking into the consideration of both SEI and PVDF consumption, which should be 55.5 wt.% (SEI) + 5.2 wt.% (PVDF) = 60.7 wt.%. Therefore, before 200°C, 60.7 wt.% of lithium was consumed. We have revised the manuscript accordingly, in page 14:

‘Note that, due to the limited amount of PVDF binder (anode composition 95 wt.% graphite, 2 wt.% PVDF, and 3 wt.% carbon black), only 5.2 wt.% of the total leached lithium is consumed by reaction (1); with another 55.5 wt.% of leached lithium consumed by the SEI layer, see Note 1 and Table 1 in the Supporting Information. Thus, 39.3 wt.% of the residual liquid lithium can exist on the graphite surface at high temperatures, leading to great potential hazards.’

6. *Should the authors add “O₂” in Figure 5b, which is coming from equation 5...?*

Response: Thank for the suggestion. The oxygen is added in the revised Figure 5, as shown in Figure R1 below, and the manuscript was revised accordingly.

Figure R1, the oxygen is added in Figure b for a better presenting of the results.

Reviewer #2 (Remarks to the Author):

*The authors reported the work entitled “In situ observation of thermal-driven SEI decomposition, gas evolution, and lithium leaching in lithiated graphite anode”. Solid-electrolyte interphase (SEI) decomposition, lithium leaching, and gas release of the lithiated graphite anode during heating were investigated by in situ synchrotron X-ray techniques and in situ mass spectroscopy. **This work discovered the critical role of SEI in anode thermal stability and the potential safety concerns of flammable gases and leached Li.** However, some issues need to be resolved before its publication in Nature Communications.*

General response: We thank the reviewer very much for his/her appreciation on the value of our work. His/her constructive comments have significantly improved the quality of our manuscript.

1. *The authors demonstrate the direct Li leaching from lithiated graphite upon heating. However, it is still unclear why and how Li leaching happens from lithiated graphite upon heating. The underlying mechanism should be clarified by combining DFT or literature.*

Response: Thanks for this valuable suggestion. We have added more theoretical discussion in the revised manuscript.

The density functional theory (DFT) calculation on the formation stability of Li-GIC (graphite intercalation compounds) at room temperature have been extensively conducted. By using 8 different DFT calculation functions, Grande *et al.* estimated the binding energy of Li-GIC is negative, as shown in Figure R2a, which is well consistent with the experimental results [*Phys. Rev. B*, **2004**, 15, 69; *Phys. Rev. B*, **2012**, 20, 85.], indicating that the lithiated graphite is stable at room temperature. With temperature change, Pande *et al.* calculated the LiC₆ phase diagram is the same as a function of temperature range from -25°C to 50°C [*Phys. Rev. Materials* **2018**, 2, 125401]. To the best of our knowledge, we do not find a direct DFT calculation on the LiC₆ formation energy change evolution at a higher temperature. Nevertheless, from the typical preparation method to make the LiC₆ (Li:C=1:5, annealing at 330°C for 24h [*J. Alloys Compd.*, **2013**, 575, 403]), we could know that the LiC₆ should be thermal dynamically favorable at 330°C. Indeed, Konar *et al.* sketched out the phase diagram of Li-GIC and shown that the LiC_x phase is thermal-dynamically stable even up to ~400°C [*Chem. Mater.*, **2015**, 27, 2566], see Figure R2b. Therefore, the existing theoretical and experiment results all indicated that the LiC₆ is thermodynamically stable at high temperatures.

Figure R2. (a) Interlayer binding energy of graphite as a function of interlayer separation calculated by LDA, GGA and five different vdW functionals in DFT calculation. Data obtained from [Z. Wang, S.M. Selbach, T. Grande, RSC Adv., 2014, 4, 3973-3983]. (b) Sketch of the Li-C phase diagram. The stage I and II Li-GIC compounds all shown thermal stable up to ~400°C. Data obtained from [S. Konar, U. Hausserman and G. Svensson, *Chem. Mater.*, 2015, 27, 2566-2575.]

However, the major difference between previous results and our new findings lies on the SEI layer, which has been ignored for a long-time when only considering the stability of chemical-prepared Li-GIC. The major contribution of our work is to clarify that the electrochemically obtained Li-GIC with SEI layers has a significantly large reaction kinetics that pure Li-GIC. With the existence of SEI surface layer, the intercalated lithium is more favorable to leach out. To further confirm the role of surface layers on determining the lithium leaching kinetics, we have introduced lithium dendrites on the graphite surface by low temperature charging in addition with the SEI layer. The results showed that the overall thermal decomposition temperature was decreased significantly. The lithium leached immediately when heating started at 25°C, the PEO oligomer peak disappeared at 43°C, with LiC₆ disappeared at 54°C, as shown in Figure R3.

At last, the DFT calculation is mainly dealing with the length scale of Angstrom and/or sub-nanometer. Due to heavy load of calculation intensity, it is thus extremely difficult to fully integrate Li-GIC, SEI layer, and/or electrolyte into a single model to calculate its dynamics. Nevertheless, as also stated in the manuscript, the thermal stability of LiC₆ itself is a good indication that by tuning the reaction kinetics with an inert SEI layer, we may can preserve the intercalated lithium from leaching out, thus leading to a safer energy storage.

[Redacted]

Figure R3. The structural evolution of dry lithiated graphite with lithium dendrites. (a) The contour plots and (d) waterfall plots of lithiated graphite anode with lithium dendrites during the heating from 25°C ~ 280°C with 2°C/min. The 2θ value has been converted into d -spacing. (b) The enlarged plots showing the SEI decomposition between 25°C to 43°C. (c) The enlarged plots of $\text{LiC}_6(100)$ reflection peak evolution of between 25°C to 54°C.

In comply with the reviewer, we have added the DFT calculation in page 3 as following: ‘In addition, by means of the density functional theory (DFT) approach, Pande *et al.* calculated that formation of LiC_6 is thermodynamically stable and not affected much with 76K temperature increasing.²² Moreover, experimental characterizations by means of differential scanning calorimetry (DSC)²³ and X-ray diffraction (XRD)¹⁷, earlier work of Drue,¹⁷ and Avdeev *et al.*²³ also suggested that the LiC_6 is thermodynamically stable up to 250°C~330°C.’

2. *It is written “charged graphite anode or charged anode” in Fig. S2a and 2b, which can confuse since the “Charged graphite anode or charged anode” can be different in full cell and half cell. It is better to use “lithiated graphite anode” to avoid confusion as always used in the whole manuscript.*

Response: We agree with the reviewer on this point. To avoid confusion, we now use the ‘lithiated graphite anode’ in the whole manuscript.

3. *The electrolyte composition of the pouch cell was missing in the materials part (Fig. S1c and text). The full name of EC and EMC is missing in SI. It is written, “PEO oligomer (-CH₂-CH₂-O)_n was reported as one of the major ingredients of the outer part of the SEI by XPS, NMR, FTIR, and DFT calculation, usually derived from ethene carbonate (EC) decomposition during the charging process of LIBs.” If it contains EMC or Dimethyl carbonate (DMC) in the electrolyte, why the authors did not*

correlate the EMC or DMC-derivation containing SEI with gas formation in the manuscript?

Response: Thanks for the suggestion. The full name of EMC is ethyl methyl carbonate, which has been added in the revised SI. The electrolyte composition is the 1.2 M LiPF₆ in ethylene carbonate (EC): ethyl methyl carbonate (EMC)=3:7 (volume ratio), which has been added in the Figure R4 below.

Figure R4. The electrolyte composition now has been added.

EC has been viewed as the most critical component for the formation of SEI on the graphite anode material. As illustrated by Kang Xu (*Chem. Rev.*, **2004**, 104, 4303-4418), ‘EC was found to form an effective protective film (SEI) on a graphitic anode that prevented any sustained electrolyte decomposition on the anode, while this protection could not be realized with propylene carbonate (PC) and the graphene structure eventually disintegrated in a process termed “exfoliation” because of PC co-intercalation.’ Therefore, the vital role of EC for the formation of stable SEI layer have been revealed compared with PC or other linear carbonate such as EMC or DMC. That is the reason why we emphasis the EC-derive SEI.

However, as an active electrolyte component, we agree with the reviewer that the role of EMC for SEI formation cannot be completely rule out. Therefore, we have modified the relevant description in page 7 of revised manuscript to ‘PEO oligomer (-CH₂-CH₂-O)_n was reported as one of the major ingredients of the outer part of the SEI by XPS, NMR, FTIR, and DFT calculation, usually derived from ethene carbonate (EC) and ethyl methyl carbonate (EMC) decomposition during the charging process of LIBs.’

4. *References are missing for PDFgui software in SI and how the PDF data in Fig.4f was obtained should be briefly described.*

Response: Thanks for the suggestions. We have added the following description and cited the relevant reference in the revised SI:

‘Synchrotron X-ray total scattering data were collected on beamline 11-ID-C at the Advanced Photo Source (APS), ANL. The rapid-acquisition PDF method was used with a wavelength of $\lambda = 0.1173 \text{ \AA}$.⁵ A PerkinElmer amorphous Si two-dimensional image-plate detector (2048×2048 pixels and 200×200 m pixel size) was used at a distance of ~ 400 mm. The two-dimensional data were converted to one-dimensional XRD data using the GSAS-II software.⁶ PDF data were obtained from Fourier transformation of the background and Compton scattering corrected data $S(Q)$ in xPDFsuite software over a Q range of $0.4\text{-}19 \text{ \AA}^{-1}$.^{7,8}

- (5) Chupas, P. J.; Qiu, X.; Hanson, J. C.; Lee, P. L.; Grey, C. P.; Billinge, S. J. Rapid-acquisition pair distribution function (RA-PDF) analysis. *J. Appl. Crystallogr.* **2003**, *36* (6), 1342-1347.
- (6) Toby, B. H.; Von Dreele, R. B. GSAS - II: the genesis of a modern open - source all purpose crystallography software package. *J. Appl. Crystallogr.* **2013**, *46* (2), 544-549.
- (7) Yang, X.; Juhas, P.; Farrow, C. L.; Billinge, S. J. xPDFsuite: an end-to-end software solution for high throughput pair distribution function transformation, visualization and analysis. *arXiv preprint arXiv:1402.3163* **2014**.
- (8) Adams, B. D.; Zheng, J.; Ren, X.; Xu, W.; Zhang, J.-G. Accurate Determination of Coulombic Efficiency for Lithium Metal Anodes and Lithium Metal Batteries. *Adv. Energy Mater.* **2018**, *8* (7).

5. *According to the International Union of Crystallography (IUCr), the correct form of indices of Bragg reflection in diffraction patterns should be without bracket. In diffraction patterns, one should call “reflection”, not “peak”. Please correct all the related parts throughout the whole manuscript.*

Response: Thanks for this professional correction. We have changed all the description to ‘reflection’ rather than ‘peak’ for XRD and have corrected the form of indices for Bragg reflection in each figure without bracket. To combine the suggestions of reviewer 3, the revised Figure 1 now as following:

Figure R5. As suggested by the reviewer, the form of the Bragg indices has been corrected in each figure.

7. There are some typos such as “degradation → degradation” in Fig. 1d, “fiction → fraction” (Fig.2).

Response: The typos have been revised, and the writing has been polished by a professional native language editor at Argonne National Laboratory.

8. Did the authors perform *in situ* HEXRD and PDF for standard materials such as NaCl or LaB6 to calibrate the temperature for each pattern, which is very important to reach each “target or marked” temperature for each diffraction or PDF pattern?

Response: Yes, we did conduct the calibration each time before measurement by using the CeO₂ powder as standard (CeO₂ NIST 647b). A typical room temperature standard and the calibration result are shown in Figure R6 as following.

Figure R6, (a) a 2D CeO₂ standard and (b) the calibrated 1D XRD reflection peaks.

For the temperature during *in situ* heating, we use a linkam THM600 furnace with precise temperature control and stability within 0.1°C. To make sure the targeted temperature was achieved, we wait extra 10 min each time after the target temperature was reached and then start to acquire the PDF data. The experimental setup with precise temperature control setups is shown as in Figure R7a. Before the experiment, the furnace was calibrated between 30°C to 350°C by using a MgO lattice parameter as standard, see Figure R7b. The different temperature reference lattice parameter for MgO were obtained from [*Physics and Chemistry of Materials*, **1999**, 27, 103-111].

Moreover, to make sure the *in situ* synchrotron X-ray experiments conducted properly, we have consulted several professional APS beamline scientists, and have them guide us during the experiments, who are also in the author list of this manuscript, as they are Dr. Yang Ren, Dr. Saul Lapidus and Dr. Wenqian Xu from beamline 11-IDC, 11-BM and 17-BM from APS of Argonne National Laboratory.

In comply with the reviewer, we have added the text and figure description for the *in situ* PDF measurement as Figure S15 in the supporting information.

Figure R7, (a)The experimental setup for the in situ PDF measurement of the lithiated graphite anode. (b) The temperature calibration data for the Linkam furnace by using a as reference at the temperature range between 30°C to 350°C .

9. *How did the authors determine the LiF and Li₂O separately in SEM (Fig.4c)?*

Response: Thanks for the question. We believe this is a misunderstanding. In Figure 4c, we did not mean the LiF and Li₂O were separately indicated by the two arrows and we agree with the review that one cannot distinguish the LiF and Li₂O from the SEM image. To avoid misunderstanding, we now changed the inset description from ‘LiF, Li₂O’ to ‘LiF and/or Li₂O’ in Figure 4c.

Figure R8, the revised figure description for Figure 4c.

10. The CO_2 intensity almost has the same order as H_2 as shown in Fig.3, what this indicates, and if it can also be correlated to something?

Response: Thanks for this valuable question. Indeed, the H_2 and CO_2 gases are two of the dominant gas species during the thermal degradation of lithiated graphite (39.3%~51% of total gas for H_2 and 22%~27.9% for CO_2 as summarized in Figure 3). By the advantages of time-resolved measurement, we revealed that the one major SEI components-PEO oligomer, accounts for the H_2 gas release, by reaction (2); while the other SEI component - lithium alkyl carbonate (ROCO_2Li , $\text{R} = \text{CH}_3\text{-}$, $\text{CH}_3\text{-CH}_2\text{-}$, $\text{CH}_3\text{-CH}_2\text{-CH}_2\text{-}$, etc.) accounts for the CO_2 generation, during the temperature range of $50^\circ\text{C}\sim 150^\circ\text{C}$, following reaction (4):

It should be noted that, in the practical Li-ion batteries, with the existence of liquid carbonate electrolyte (EC, EMC), the portion of CO_2 may be even higher than in the case of dry anode. For example, Baird *et al.* have summarized the reported battery vent gas composition during thermal failure, the CO_2 can account for from 10% up to 95% of volume fraction of the total gas, as shown in Figure R9.

At last, compared with the flammable and explosive properties of H_2 , the CO_2 shows much less safety concern, since it is inert at most conditions. The only safety concern on CO_2 is it can cause the battery swelling or eruption.

In comply with the reviewer, we have added more discussion on the effect of CO_2 in the revised manuscript in page 13: ‘Oppositely, the CO_2 ratio is lower than that in the full batteries, indicating the major generation of CO_2 come from the decomposition of electrolyte and cathode. By the advantages of time-resolved measurement, we revealed

that the one major SEI component-PEO oligomer, accounts for the initial H₂ gas release, by reaction (2); while the other SEI component-lithium alkyl carbonate (ROCO₂Li) accounts for the CO₂ generation, following the reaction (4).’

Figure R9. Battery vent gas species composition from literature. Figure obtained from [*J. Power Sources*, 2020, 15, 227257]

Reviewer #3

The manuscript "In situ observation of thermal-driven SEI decomposition, gas evolution, and lithium leaching in lithiated graphite anode" by Xiang Liu, Khalil Amine and coworkers reports interesting results on the thermal decomposition of graphite LIB anodes. There have been thermal degradation experiments before but still the topic is relevant to safety and, still, the thermal degradation pathway and detailed safety hazards of lithiated graphite remain elusive. The decomposition reactions are complex indeed. Therefore, in particular the here presented combination of online gas analysis and in-situ x-ray diffraction is to my knowledge new and important. The authors can deduce new conclusions from their approach via the direct correlations, and they can, as they state, for the first time directly observe the thermal expansion and breakdown of the PEO oligomer.

Indeed, the study is timely and important and deserves publication in Nature Communications, I recommend after minor revisions. As a minor remark, I recommend that the authors rework some of their figures and correct the English as they use some slightly awkward expressions.

Response: Thanks for your appreciation on the importance of our new finding to the battery community. We have revised the manuscript accordingly.

- 1. As a first suggestion I would remove the contour plot of Figure 1a to the Supporting Information as they did for Figure 6 as well. This should be done because I absolutely think that a basic description of the setup needs to be part of the main manuscript (the authors start with the results first sentence referring to SI Figures S1 and S2 for a detailed description of lithiated anode sample preparation and the experimental setups). Also Figures 1c and d could be combined showing a few curves only for clarity and instead add a plot on, for instance, peak maximum with temperature to better show the onset of degradation. The message is a bit lost in too many curves at too many temperatures.*

Response: Thanks for the nice suggestion to make our manuscript more readable. We have made the correction accordingly, the revised Figure 1 is now shown as following. And we have moved the contour plots to the support information part. The Figure 1c and 1d now combined as one figure to show less lines with more obvious change, see below in Figure R10c.

Figure R10. (a) The scheme of the experimental setups. (b) The waterfall plots of lithiated graphite anode during heating from 25°C to 280°C with 2°C/min. The 2θ value has been converted into d-spacing following Bragg's rule for a better comparison to the results using different X-ray sources. (c) The PEO oligomer lattice expansion between 25°C~40°C, with a coefficient of thermal expansion (CTE) of $167 \times 10^{-6}/^\circ\text{C}$, then PEO degradation during 40°C~60°C.

2. Should be "charge/discharge" in the following sentence: During the discharge/charge process, Li^+ ions are inserted/extracted between the graphene planes without significantly disturbing the graphite host structure, thereby achieving reversible (de)intercalations.

Response: Thanks for this careful correction. Indeed, in a full battery, the charge/discharge is correlated to the process of lithium-ion insertion/extraction into/from the graphite anode. We have modified this sentence.

3. What is phase fiction in Figure 2? Do you mean "fraction"?

Fig.2: a/b and c/d are exchanged in caption. Please proof read!

Response: Thanks for the correction. The “friction” should be “fraction”. We now have modified the Figure 2. We have checked the whole manuscript, and also have an Argonne professional language editor to improve the writing.

4. Fig.4d: What is an "induction" experiment? ... I guess the authors mean an O₂ exposure experiment? Also better replace to: Influence of oxygen on lithiated anode... Induce/induction means something else.

Response: Thanks for the suggestion. We agree with the reviewer on this point and to avoid confusion, we now have changed the ‘oxygen induction experiment’ to ‘the oxygen exposure experiment’ in the revised manuscript accordingly.

For example, in page 6, ‘DSC characterization with oxygen exposure experiment further proved the existence of highly reactive residual lithium’ and page 14 ‘As shown, with oxygen exposure, an immediate heat generation peak was triggered, with 123 J/g, 795 J/g, and 852 J/g at 200°C, 240°C, and 260°C, respectively.’

5. What is in Fig.5 "limited Lithium"? I guess you mean a limited amount of Li? Then write it: "Small amount of residual lithium" in same style like Figure next to it.

Response: Thanks for the correction. We have changed the figure description accordingly, see Figure R11 below.

Figure R11. The revised Figure 5, change “limited lithium” to “small amount of lithium”.

Reviewer #1 (Remarks to the Author):

The authors have addressed all my concerns. I am satisfied with the revisions. I recommend accepting at the present state.

Reviewer #2 (Remarks to the Author):

The authors have resolved most of my concerns, however some of them are not fully addressed concerning my 2nd and 5th previous comments.

1. The text of “charged graphite anode or charged anode” in Fig. S2a and 2b has not been corrected to “lithiated graphite anode”.
2. Regarding the correct form of indices of Bragg reflection in diffraction patterns and “reflection”, some of them are still there, such as in Legends of Fig.2, Fig. S7, S8, S14, and S18, as well as in text of main manuscript and SI.
3. In addition, “In situ” should be in italic form. Both normal and italic forms are used in this work.

Reviewer #1 (Remarks to the Author):

The authors have addressed all my concerns. I am satisfied with the revisions. I recommend accepting at the present state.

The authors appreciate reviewer's recommendation to publish.

Reviewer #2 (Remarks to the Author):

The authors have resolved most of my concerns, however some of them are not fully addressed concerning my 2nd and 5th previous comments.

The authors thank the reviewer for the careful evaluation. We have revised our manuscript accordingly.

1. The text of “charged graphite anode or charged anode” in Fig. S2a and 2b has not been corrected to “lithiated graphite anode”.

Response: We have revised the Figure S2a and 2b by using the term of ‘lithiated graphite anode’.

2. Regarding the correct form of indices of Bragg reflection in diffraction patterns and “reflection”, some of them are still there, such as in Legends of Fig.2, Fig. S7, S8, S14, and S18, as well as in text of main manuscript and SI.

Response: Thanks for the correction. We now have revised the term of ‘reflection’ for the legends of Figure 2, Figure S7, S8, S14 and S18, also in the main text as on page 8-9 and for supplementary information on page 9.

3. In addition, “In situ” should be in italic form. Both normal and italic forms are used in this work.

Response: Thanks for the suggestion. We now modified all the ‘In situ’ in main text and SI with the right format.